# Clinical Findings of *Listeria monocytogenes* Infections with a Special Focus on Bone Localizations

**DOI:** 10.3390/microorganisms12010178

**Published:** 2024-01-16

**Authors:** Marco Bongiovanni, Claudio Cavallo, Beatrice Barda, Lukasz Strulak, Enos Bernasconi, Andrea Cardia

**Affiliations:** 1Division of Infectious Diseases, Ente Ospedaliero Cantonale, 6900 Lugano, Switzerland; beatrice.barda@eoc.ch (B.B.); enos.bernasconi@eoc.ch (E.B.); 2Division of Neurosurgery, Ente Ospedaliero Cantonale, 6900 Lugano, Switzerland; claudio.cavallo@eoc.ch (C.C.); lukasz.strulak@eoc.ch (L.S.);

**Keywords:** *Listeria monocytogenes*, bone infections, spondylodiscitis

## Abstract

*Listeria monocytogenes* is a Gram-positive pathogenic bacterium which can be found in soil or water. Infection with the microorganism can occur after ingestion of contaminated food products. Small and large outbreaks of listeriosis have been described in the past. *L. monocytogenes* can cause a number of different clinical syndromes, most frequently sepsis, meningitis, and rhombencephalitis, particularly in immunocompromised hosts. *L. monocytogenes* systemic infections can develop following tissue penetration across the gastrointestinal tract or to hematogenous spread to sterile sites, possibly evolving towards bacteremia. *L. monocytogenes* only rarely causes bone or joint infections, usually in the context of prosthetic material that can provide a site for bacterial seeding. We describe here the clinical findings of invasive listeriosis, mainly focusing on the diagnosis, clinical management, and treatment of bone and vertebral infections occurring in the context of invasive listeriosis.

## 1. Introduction

*Listeria monocytogenes* is a Gram-positive, motile, facultative, anaerobe bacteria that inhibits a broad ecologic niche [1,2]. The microorganism can be isolated from soil, water, and vegetation, including raw vegetables intended for human consumption without further processing [3,4]. Newer chromogenic media may offer advantages in the detection of contaminated food [5,6]. The surface contamination of meat and vegetables is common, with up to 15% of these foods harboring the microorganism. Furthermore, *L. monocytogenes* is a transient inhabitant of both animal and human gastrointestinal tracts; intermittent carriage suggests possible frequent exposure [7,8]. Usually, the gut is the source for microorganisms in case of invasive listeriosis; the virulence factor ActA is associated with carriage development [9]. The microorganism has a competitive advantage against other Gram-positive and Gram-negative bacteria in cold environments, such as refrigerators; it is also amplified in spoiled food products, possibly leading to increased alkalinity. Feeding of spoiled silage with a high pH has resulted in epidemics of listeriosis in sheep and cattle [10]. Several foodborne outbreaks of listeriosis have been the result of animal epidemics; the first one occurred in Canada and was associated with the ingestion of contaminated coleslaw [11]. Subsequently, many other foodstuffs have been implicated in different outbreaks, including cheeses made with raw or pasteurized milk or milk derivates [12,13,14,15,16,17,18,19,20,21,22,23], meat products [24,25,26,27,28,29], and fruits and vegetables [30,31,32,33,34]. Table 1 summarizes the food products that are usually implicated in the occurrence of foodborne listeriosis. In addition, hospitalized individuals also seem to be at risk of acquiring *L. monocytogenes* infections [35]. To optimize the tracking of listeriosis cases, whole-genome sequencing has been developed; this has replaced older techniques, such as serotyping [36,37]. However, the question of why outbreaks of listeriosis can occur in humans remains incompletely understood; a possible enhancement of organism-specific virulence factors may play a role in developing epidemic dissemination.

Also, sporadic cases of listeriosis can be foodborne-related; reports of sporadic cases of *L. monocytogenes* infection in the absence of documented outbreaks have been associated with different food products that could represent vehicles for the occurrence of sporadic invasive listeriosis in humans [38]. Consequently, *L. monocytogenes* can be considered a common contaminant of food products; the ingestion of small quantities of this microorganism occurs frequently in humans [39]. *L. monocytogenes* usually grows in biofilms or in food products that have not undergone pasteurization and those which are kept at cold temperatures. Invasive diseases occur when the ingestion of a large number of microorganisms overwhelm the innate host-defense systems at the gastrointestinal, liver, or spleen levels. Although the annual rate of sporadic listeriosis in Europe and North America is usually <1/100,000 population per year [40,41,42], the development of this infection is associated with a high burden of costs [43,44]. Sporadic listeriosis usually follows seasonal variations, being more common during the spring and summer seasons; it is mainly associated with the increased consumption of higher-risk products during warmer periods. In addition, the risk of developing invasive listeriosis could be associated with the presence of pre-existing damage in the gastrointestinal mucosa due to other microorganisms that induce viral gastroenteritis and that have seasonal patterns that may overlap with those of listeriosis. Such damage may allow the translocation of *L. monocytogenes* from the gastrointestinal tract, followed by the subsequent development of invasive diseases [45].

All isolates of *L. monocytogenes* are able to produce all the virulence factors characteristic of the species. Several virulence factors of *L. monocytogenes* have been identified and extensively characterized at both the molecular and cell biologic levels, including the hemolysin (listeriolysin O), two distinct phospholipases, a protein (ActA), several internalins, and others. Study of these provides an impressive improvement in our knowledge of the mechanisms used by *L. monocytogenes* to interact with mammalian host cells, escape the host cell’s killing mechanisms, and spread from one infected cell to others. In particular, after the infection of the host cells, the microorganisms are internalized in a vacuole. Expression of listeriolysin O promotes escape from the vacuole into cytoplasm, where *L. monocytogenes* can replicate. The intracytoplasmic microorganisms use the actin of the host cell, in conjunction with their ActA protein, to promote their motility intracellularly, their location in protruding pseudopods, and the engulfment of the pseudopods by the adjacent host cells. After their uptake by adjacent cells, the bacteria escape the now-double-membrane-bound vacuole by means of listeriolysin O and the phospholipases, and the cycle continues to repeat. The production of listeriolysin O in the host cell is under stringent regulatory control; this virulence factor is expressed in the vacuole but not in the cytosol, preventing the killing of the host cell and allowing the host cell cytoplasm to serve as a safe haven for microorganism survival and replication. In addition, several molecular subtyping tools have been developed to facilitate the detection of different strain types and lineages of the pathogen, including those implicated in common-source outbreaks of the disease [46]. Genetic studies continue to identify genes essential for the virulence of *L. monocytogenes* [47,48]. Although the key virulence factors known to date (listeriolysin O, phospholipases, ActA, internalins A and B, and others) are present in all serotypes, their regulation of expression may differ among the serotypes. Furthermore, strains of serotypes 1/2b and 4b may have additional virulence determinants, which will not become identified until and unless strains of these serotypes become included in genetic studies of virulence. The identification of potentially unique virulence factors of these strains may be aided by the use of comparative genomic analyses.

Host-specific conditions also contribute to the increased risk of invasive listeriosis [49,50]. In particular, cases of invasive listeriosis are most commonly described in the first month of life or among elderly individuals. The fetus is mainly infected during maternal sepsis or secondary–peri-vaginal or peri-anal colonization in the birthing parent, with transmission occurring through the birth canal. Infants usually do not have an adequate host defense, mainly in cases of impairment of macrophage and cell-mediated immune function; therefore, invasive listeriosis can easily develop in case of colonization of the liver, respiratory tract, or gastrointestinal tract. Pregnant people have usually a decreased gastrointestinal motility and also a slight impairment in their cell-mediated immune response to *L. monocytogenes*; both these conditions may predispose them to invasive listeriosis, followed by transplacental infection of the infant [51,52,53,54]; this can finally lead to the delivery of a premature and often severely ill newborn. Spontaneous recovery of the birthing parent from invasive listeriosis normally occurs after the delivery; the administration of specific and appropriate antibiotic therapy can improve the prognosis of the infant and also accelerate the clinical recovery of the birthing parent. When the infant is infected through a colonized birth canal, clinical disease in the infant usually develops 7–14 days later. A direct cutaneous invasion is unlikely in this context; aspiration of *L. monocytogenes* into the respiratory tract or by swallowing the microorganism can occur only during the incubation period. Recently, a unique outbreak of neonatal listeriosis has been described. *L. monocytogenes* was spread through contaminated mineral oil used to clean infants after delivery from healthy birthing parents, with cross-contamination of shared mineral oil; the index case was infected through the placental route of maternal–fetal infection [55].

The increased risk of invasive listeriosis among the elderly usually reflects the increasing incidence of other immunosuppressive conditions in this specific population, such as solid or hematological malignancies, chronic diseases leading to immunological impairment such as diabetes or renal failure, or immunosuppressive treatments. In particular, malignancies may lead to abnormalities in the gastrointestinal mucosa and the impairment of effective macrophage function in liver, spleen, and peritoneum, both directly or secondary to chemotherapy- or radiation-induced damages; finally, bacterial translocation from the gastrointestinal tract is favored. The increasing use of immunosuppressive treatments with a specific effect on cell-mediated immune function as corticosteroids or cyclosporine A, as well as the use of biologic treatments with an immune modulator effect, such as tumor necrosis factor-alpha inhibitors, can also contribute to an increased risk of invasive listeriosis [56,57,58].

Among the causes of immunosuppression, HIV infection has been linked to the occurrence of sporadic invasive listeriosis [59]. In particular, earlier studies have described a 500–1000-fold greater risk of developing invasive listeriosis among HIV-infected individuals compared to the general population. Subsequently, a progressive reduction in reported cases has been observed, due to dietary recommendations to prevent foodborne illnesses and, above all, due to the wide use of trimethoprim-sulfamethoxazole as *Pneumocystis jirovecii* pneumonia prophylaxis, which *L. monocytogenes* is also susceptible to; furthermore, a possible contribution to the reduction in cases may be secondary to the widespread use of more efficient antiretroviral treatments that induce a restoration of immune system function [60].

*L. monocytogenes* is, overall, considered to be one of the most important foodborne pathogens associated with the occurrence of febrile gastroenteritis outbreaks. Several foods have been described as vehicles of these outbreaks, including fresh cheese, ready-to-eat meat, shrimps, rice or corn salad, and chocolate milk [15,22,61,62,63,64,65,66]. In these outbreaks, symptoms develop soon after ingestion (approximately 24 h) and attack rates have been found to be significantly greater when compared to invasive listeriosis. These high attack rates are not usually related to the enhanced intrinsic virulence of the *L. monocytogenes* strain but rather to the heavy contamination of the ingested food.

A reduction in the overall incidence of listeriosis could be due to a larger promotion of dietary recommendations to high-risk individuals, including pregnant people, people with malignancies, or people undergoing transplantation [67]. It is more likely that this reduction is due to the worldwide increase in awareness in the food-processing industry, including hazard analysis at critical control point (HACCP) [68,69]; above all, the improvement may be attributable to programs that have been formed to reduce food contamination with different microorganisms including *L. monocytogenes*, *Salmonella* spp., *Escherichia coli*, and *Campylobacter* spp. [70,71,72]. These activities have provided augmented protection for fresh, unprocessed food products that may not have been cooked or pasteurized and that are at higher risk of conveying foodborne illnesses. In addition to hazard analysis, regulatory agencies have implemented the control of those microorganisms that have the ability to potentially contaminate food. The U.S. Food and Drug Administration has developed strict recommendations for the control of *L. monocytogenes* in the food industry [73], mainly including the use of whole-genome sequences (WGSs) [74]. Recently, Conrad et al. [75] described how, starting from five cases of invasive listeriosis in Kansas, the use of WGSs enabled the identification of *L. monocytogenes* contamination of ice cream products in three other states; the facilities where the ice creams were being produced were located in Texas and Oklahoma, suggesting long-standing contamination. Other countries have adopted less stringent guidelines, allowing a small amount of contamination (<10^2^ CFU/g); this small permitted amount was chosen to enable a balance to be attained between the protection of public health and the needless condemnation of otherwise edible food products. While invasive listeriosis seems more common in some European countries than in United States, it is still unclear whether these differences can be attributed to the less stringent standards in Europe. It remains therefore debatable if a “zero tolerance” approach for *L. monocytogenes* contamination of food could be preferable to a risk assessment approach [76].

## 2. Clinical Findings Due to *Listeria monocytogenes*

*L. monocytogenes* infections are associated with a broad variety of clinical findings in both humans and animals. Sepsis caused by *L. monocytogenes* was first described in an epizootic affecting South African rodents and in laboratory colonies of rabbits [77,78]. The species name monocytogenes was suggested as a result of the production of monocytosis in blood; while monocytosis-producing antigens have been considered to be a virulence factor for *L. monocytogenes*, monocytosis in the peripheral blood is not considered to be a distinguishing factor in finding human infections [79].

Many wild and domestic animals are susceptible to invasive listeriosis. Animals can acquire *L. monocytogenes* through grazing; this is further amplified by fecal contamination in soil and vegetation. However, farm animals and in particular ruminants can acquire listeriosis not only from forage, but also from silage. In fact, a very intense keratoconjunctivitis which farmers call “silage-eye” is often described among sheep and goats affected by listeriosis; the condition has been correlated with the intake of silage that is not well fermented and is therefore potentially contaminated by *L. monocytogenes*. In ruminants, *L. monocytogenes* has been implicated as a possible cause of abortion and prematurity [80]. The clinical syndromes associated with listeriosis in humans were discovered later. Neonatal listeriosis was first described in Europe among premature septic newborns during the post-war period [81]; subsequently, other reports described neonatal meningitis as late-onset listeriosis occurring in the post-partum period. In the developed world, listeriosis is a frequent cause of neonatal meningitis; however, the wide use of antibiotic prophylaxis to prevent group B streptococcal infection has reduced cases of neonatal listeriosis [82,83].

## 3. Bone and Vertebral Infections by *L. monocytogenes*

*L. monocytogenes* only rarely causes bone and joint infections; this usually occurs in the context of prosthetic material that can provide a site for bacterial seeding.

### 3.1. Imaging Techniques

When suspecting bone or vertebral infections, the use of imaging techniques such as radiography or CT scans could provide valuable information in terms of bone erosions and vertebral bone integrity, mainly in the later stages of the disease; during the early stage of infection, no significant finding is usually detected. Furthermore, spinal stability must be assessed among patients in whom surgical management is being considered. Indeed, vertebral collapse, kyphotic deformity, and loss of normal lordosis can be found in advanced infections. CT also provides guidance for percutaneous aspirations in order to provide specimens for bacteriologic analysis in the presence of a fluid collection. MRI is the gold standard and represents the diagnostic imaging modality of choice. It should be performed among all patients for whom a spinal infection is suspected, unless contraindicated. Unenhanced T1-weighted images usually reveal a hypointense signal at the level of the end plates in the vertebral body and loss of a normal hyperintense fat signal in the vertebral bone marrow. T2-weighted imaging reveals a high signal corresponding to edema in the disk space and occasionally in the bone and paravertebral soft tissues. Gadolinium-enhanced T1-weighted imaging can demonstrate the contrast enhancement of the vertebral body, end plates, the prevertebral and paravertebral soft tissues, and the epidural space. Whenever the MRI is contraindicated or non-diagnostic (e.g., due to the presence of metallic implants causing artifacts), other imaging modalities should be considered. CT myelography provides another way of visualizing the spinal cord and ruling out compression in the setting of suspected cauda equina syndrome. On the contrary, nuclear medicine scans with radionuclide studies offer a high degree of sensitivity in the early stages of the disease. Spinal infections can occasionally be multifocal, so the whole spine should be scanned if an infectious focus is detected.

### 3.2. Microbiological Diagnosis

The determination of a microbiological diagnosis of *L. monocytogenes* bone or vertebral infection is challenging, especially in the absence of referred exposures or negative blood tests. In this context, aspiration biopsy or surgical sampling represent the optimal method of providing a valid microbiological diagnosis. As a consequence, empiric antibiotic therapy should be delayed if the patient is hemodynamically stable and has no neurological signs in order to obtain valid samples for cultures; postponing antimicrobial administration can improve the microbiological yield, so it could be preferably deferred in the absence of life-threatening conditions or spinal cord involvement [84]. However, the initiation of an antibiotic treatment does not always preclude undertaking a biopsy [85]; in those cases where antibiotic treatment has already been started, it has been demonstrated that interrupting and withholding antibiotics for 2 weeks led to a better yield compared to holding for only 3 days pre-biopsy [86]. These data can vary according to the pharmacokinetics, dose, duration, and bone penetration of the selected antibiotic. Nevertheless, a short duration of empiric antibiotic exposure does not negatively impact pathogen recovery and therefore is not an absolute contraindication for biopsy [87]. Therefore, all these diagnostic and therapeutic issues should be taken into consideration when managing *L. monocytogenes* vertebral infections.

### 3.3. Surgical Approach

In the absence of neurological deficits or sepsis, the optimal therapeutic approach comprises medical management with adequate intravenous antibiotics and immobilization of the affected spinal segment. Antibiotic therapy should be started as soon as the microorganism has been isolated in order to achieve sterilization of the infected bone or vertebral disc and prevent the occurrence of a neurological deficit or painful deformity. The duration of antibiotic therapy varies depending on the extent of bone involvement and the status of the patient’s immune system. Neurosurgical intervention should be considered only after taking into account a given patient’s neurological status as well as the extent of bone erosion and the specific vertebral level involved. The principles of surgical treatment include debridement of infected tissue, decompression of neural elements, and the restoration of spinal alignment and/or correction of spinal instability. The presence of neurological deficits is considered to be the most important factor in the decision-making process. Regardless of the duration of the weakness, emergency surgical intervention is offered unless the motor deficit is minimal. Patients for whom non-surgical management is considered should be carefully monitored; this is because early progression with neurological deterioration may occur rapidly. Surgical approaches for spinal infections are usually dictated by the site of compression (ventrally vs. dorsally located lesions) and tailored to the vertebral level involved. The nature of the compressive lesion is also relevant; the liquid collection of pus can be drained, whereas a mass of granulation tissue or retro pulsed bone are better addressed with an open surgical approach. In addition, the optimal surgical approach is selected after consideration of the intrinsic features of each anatomic region of the spine and the likelihood of postoperative instability. The degree of kyphotic deformity, the number of vertebral elements involved, and the bone and posterior tension band integrity can be used to determine the extent of spinal instrumentation required to restore stability. Surgical intervention is also indicated after the failure of medical management or patients with chronic pain, significant deformity, or spine instability in the setting of spinal infection or its sequelae.

### 3.4. Antibiotic Treatment of Bone and Vertebral L. monocytogenes Infections

Reports of *L. monocytogenes* bone infections are usually described in patients with predisposing factors, such as diabetes, leukemia, or receipt of long-term corticosteroids or immune-modulant treatments [88,89,90,91,92]. Usually, native vertebral *L. monocytogenes* infections have an insidious course, with symptoms, especially back pain, that could be present for over a year, as described in previous reports [93,94]. In the review by Charlier et al., more than 70% of cases of listeriosis involving bone and joint infections were subacute or chronic at the onset. Furthermore, most of these cases occurred in the hip (60%) and in prosthetic joints [88]. In this review, patients with osteomyelitis caused by *L. monocytogenes* showed only a mild increased in inflammatory markers compared with those with other forms of bacterial osteomyelitis. Vertebral osteomyelitis represents an even less frequent localization of invasive listeriosis. To date, eight cases [93,94,95,96,97,98,99,100] have been reported in the literature and most of them had significant risk factors for developing invasive listeriosis (Table 2). Figure 1 showed the MRI evolution of a patient with epidural abscess by *L. monocytogenes* from diagnosis to complete recovery after both surgical and antibiotic treatment [101]. All these patients were treated with ampicillin or amoxicillin or benzyl penicillin; five patients received a treatment combination with aminoglycosides; treatment duration was highly heterogeneous among these reports, ranging from 6 to 28 weeks in accordance with possible delayed clinical responses. Only one report [97] used trimetoprim/sulphametoxazole as an oral maintenance treatment; however, this was not combined with amoxicillin in any of the studies. Oral use of amoxicillin was described in two other reports and was administered for a total of 12 and 18 weeks, respectively [98,99]. The antibiotic treatment should often be associated with surgical intervention in cases of spinal *L. monocytogenes* infections, especially for those patients experiencing neurological deficit, cord compression, destruction of the vertebrae with instability, large epidural abscesses, or inadequate responses to antimicrobials [102].

## 4. Other Clinical Features

### 4.1. Listeriosis and Pregnancy

Pregnant people have an increased risk of *L. monocytogenes* infections; these can lead to chorioamnionitis and finally to early-onset neonatal listeriosis [51]. Infants with listeriosis have a peculiar constellation of clinical features, including prematurity sepsis at birth, fever, cutaneous maculo-papular exanthema, and jaundice [103]. In this context, the mortality rate is very high, even when a prompt, specific antibiotic treatment is started. Autopsy findings showed chorioamnionitis in placental remnants and multiple granulomas in the spleen and liver of the infants; the syndrome, when first described, was therefore called “granulomatosis infantispetica” [81]. While such infants usually have dramatic findings, their birthing parents may be asymptomatic or may commonly have only mild symptoms, like flu-like or urinary or gastrointestinal symptoms, before their blood cultures were shown to be positive for *L. monocytogenes*. The rapid administration of antibiotic treatment to birthing parents with *L. monocytogenes* bacteremia can prevent transplacental infection, with a delivery of an uninfected infant [104]. However, this is a very unusual condition and can only happen when a community-based outbreak of *L. monocytogenes* is identified in specific geographical areas through public health alerts. Symptoms of late-onset neonatal meningitis due to *L. monocytogenes* usually occur 1–2 weeks after delivery; these symptoms have been found to include fever, irritability, bulging fontanelle, and meningismus [105]. In this context, the birthing parent does not develop any septic complication during pregnancy, delivery, or the post-partum period.

### 4.2. Meningoencephalitis

Invasive *L. monocytogenes* infections have been associated with meningitis in adults. Usually, the clinical symptoms are those of subacute bacterial meningitis with fever, headache, and neck stiffness that can develop over several days [106]. During epidemics of foodborne listeriosis, meningitis caused by *L. monocytogenes* can also occur in apparently healthy individuals of all ages; on the contrary, in sporadic diseases, a defect in cell-mediated immune function can predispose subjects to this finding of invasive listeriosis. In addition, *L. monocytogenes* can induce rhomboencephalitis in humans and in animals; this is described mainly as circling disease [107]. When these features appear, fever, headache, nausea, and vomiting occur early, and signs of meningeal irritation are less commonly observed. Subsequently, multiple abnormalities of cranial nerves develop with associated cerebellar dysfunction, mainly ataxia. Fever is not present in up to 15% of patients, leading to a more difficult diagnosis. However, the presence of micro abscesses in the cerebellum and diencephalon could aid in the diagnostic process. This variant has a mortality of 50% and a high risk of neurological sequelae, despite prompt administration of antibiotic treatment.

It has been reported that *L. monocytogenes* infiltrates the brain either directly from blood or the nervous cell fibers connected to peripheral tissues [108]. *L. monocytogenes* is able to cross the blood–brain barrier, recognizing specific receptors as E-cadherin or Met at the surface of the barrier, and then adheres by using Internalin- (Inl) A and Inl-B and finally passes the blood–brain barrier [109]. Another way in which *L. monocytogenes* enters the brain is the transportation of the microorganism inside infected phagocytic cells, such as monocytes and dendritic cells or bone marrow myelomonocytic cells, directly to the CNS (the so-called Trojan horse mechanism) [110,111]. Afterwards, the bacterium is able to infect the brain neurons spreading from these carrying cells [112]. Recently, it has been demonstrated in a mouse model of neurolisteriosis using hypervirulent *L. monocytogenes* strains that monocytes are necessary and sufficient to induce neuroinvasion. It has also been documented that Inl-B possesses a major role in this process, whereas Inl-A is not involved [113].

### 4.3. Listeria monocytogenes Sepsis

Bacteremia by *L. monocytogenes* without central nervous system involvement represents approximately one-third of adult cases of invasive listeriosis. Symptoms are usually a-specific, but fever and chills are often present. The occurrence of *L. monocytogenes* sepsis is often associated with pre-existing cancers, organ transplants, or other causes of immune depression, and has a mortality rate of up to 30%. In this context, the symptoms are nonspecific and can mimic the sepsis caused by other Gram-positive or Gram-negative bacteria [114,115].

### 4.4. Gastroenteritis

*L. monocytogenes* can cause febrile gastroenteritis with diarrhea and abdominal pain, especially during large outbreaks of foodborne adult listeriosis, with a high burden of microorganisms in the contaminated food [15,22,23,61,62,63,64,65,116]. Most patients are healthy before the development of the infection; bacteremia is an unusual finding in this setting and most patients develop symptoms within 24 h following exposure; a large amount of microorganisms (up to 10^9^ CFU/g) is usually found in the contaminated food. During these outbreaks, pregnant people have a particularly high risk of developing sepsis and invasive listeriosis; the isolation of *L. monocytogenes* from stool is unusual, but serological tests have been widely used to better define the extent of the outbreaks. In the outbreaks reported to date, reported foods have been identified as shrimp salad, chocolate milk, corn, deli meats, and fresh cheese. Invasive listeriosis with meningoencephalitis could finally occur in this context when the gastrointestinal mucosa lost its integrity for other bacterial or viral concomitant gastroenteritis [49,117].

### 4.5. Endocarditis

Endocarditis caused by *L. monocytogenes* usually follows transient bacteremia from a gastrointestinal focus with the subsequent endovascular infection of an abnormal heart valve; over 50% of cases of endocarditis cause by *L. monocytogenes* involved prosthetic valves, whereas cases in native valves are sporadic [118,119]. Diagnostic criteria included the presence of a prosthetic valve with or without vegetation and a continuous bacteremia by *L. monocytogenes*; septic emboli and abscesses in other organs are relatively frequent and can occur in approximately two-thirds of patients. Aortic and mitral valves are the most commonly involved valves. In native valve endocarditis, *L. monocytogenes* infection can sporadically follow previous episodes of streptococcal bacterial endocarditis or other valvular heart diseases. Reports of patients with malignancy, diabetes, prolonged steroid therapy, and renal and liver transplantation with *L. monocytogenes* endocarditis have been published to date [120]. The clinical presentation is usually nonspecific for *L. monocytogenes* and includes prolonged fever, chills, and, ultimately, signs of congestive heart failure. In these cases, diagnosis can be obtained only by systematically performing blood cultures. *L. monocytogenes* can also cause arterial infections that involve prosthetic abdominal and aortic grafts or native abdominal aortic aneurysms [121]. The mortality rate of this condition approached 40% before 1985 but has been reduced to 12% with better recognition and surgical management. In this context, a multidisciplinary approach is mandatory for better management of the antibiotic treatment, the surgical intervention, and the increased risk of systemic complications.

### 4.6. Abdominal Infections

*L. monocytogenes*-associated hepatitis has been described in several case reports [122]. Though the diagnosis is often unsuspected, severe diseases can occur; autopsy findings showed micro abscesses and occasional granulomas that were similar to those observed in neonatal disease [123]. Solitary and multiple liver abscesses with fever have also been reported [124,125]. Predisposing factors for liver complications caused by *L. monocytogenes* included cirrhosis, liver transplantation, diabetes mellitus, and alcoholism.

Recently, *L. monocytogenes* has been described as a possible cause of biliary tract infections, mainly by retrograde infection from contaminated food and because the microorganisms are resistant to bile [126,127]. Immunosuppression due to corticosteroids and use of biologic agents to treat underlying conditions are well-established risk factors for this infection; the mortality rate is high, mainly due to inappropriate antibiotic therapy decisions following misleading diagnoses.

*L. monocytogenes* can also cause isolated episodes of peritonitis, especially among patients receiving peritoneal dyalisis with isolation of the microorganism from dialysate or blood cultures, or among those with advanced liver diseases [128,129,130,131,132]. Infections are usually secondary to the translocation of the microorganism from the gastrointestinal tract among patients who have ingested *L. monocytogenes* with food. The mortality rate is low; it is comparable to that of spontaneous bacterial peritonitis caused by other microorganisms.

### 4.7. Cutaneous Infections

Cutaneous listeriosis is an occupational hazard for veterinary workers exposed to infected amniotic fluid or placental remnants that are removed from the birth canal of animals [133,134]. Also, conjunctivitis has been reported among laboratory workers [135]. In these conditions, *L. monocytogenes* is usually isolated by the multiple papulo-pustular lesions of the skin; findings are similar to those observed in infants with early-onset disseminated listeriosis. In adults, the infection is usually self-limited and people recover spontaneously without antibiotic treatment; however, its occurrence is easily preventable with the appropriate use of gloves and other protective wears.

## 5. Conclusions

In this paper, we present a comprehensive description of clinical findings surrounding *L. monocytogenes*. We mainly focused on bone and vertebral infections caused by *L. monocytogenes*, because these localizations are usually underestimated, except in cases of outbreaks. In fact, in the literature, very few data have been reported for this specific localization, and most of them are extrapolated by case reports. We described the diagnostic and the clinical management of patients with bone infections due to *L. monocytogenes*, as well as the surgical and the optimal antibiotic treatment for this condition. Despite the limited number of reports, consideration for *L. monocytogenes*-associated osteomyelitis should be taken in account as part of the differential diagnosis, even in the absence of prosthetic material, in the context of epidemiologic risk factors especially. However, this diagnosis should also be considered in those individuals living in areas with relatively low incidences of *L. monocytogenes* infections, because sporadic outbreaks can occur everywhere.

## Figures and Tables

**Figure 1 microorganisms-12-00178-f001:**
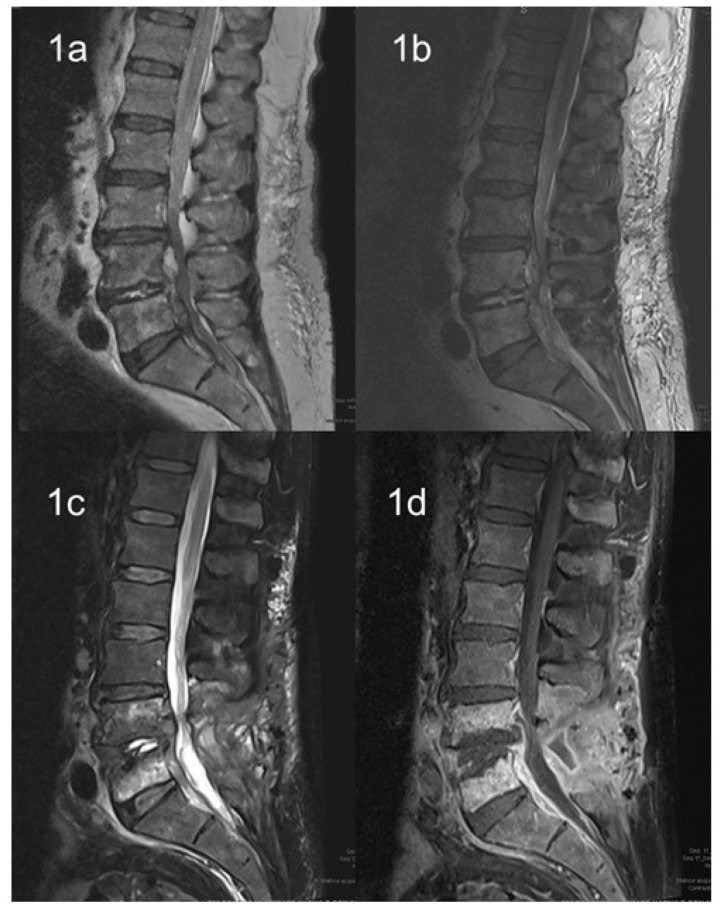
Evolution of MRI in a patient with epidural abscess by *L. monocytogenes*. (**1a**) T2 weighted image at baseline; (**1b**) T2 weighted image after 10 days; (**1c**,**1d**) T2 and T1 weighted with contrast at clinical resolution.

**Table 1 microorganisms-12-00178-t001:** Foods that are usually implicated in foodborne listeriosis.

Dairy Products	Fruits and Vegetables	Meat Products	Fish Products
Pasteurized whole milk Chocolate milk Soft cheese (different types) Hard cheese Mexican-style cheese Goat cheese Ice cream Fresh cream	Coleslaw (cabbage) Lettuce Corn Rice salad Salted mushrooms Sprouts Strawberries Nectarines Apples Cantaloupes Blueberries Stone fruit	Delicatessen foods (deli meats) Pâté Foie gras Uncooked hot dogs “Rillettes” Pork tongue in aspic Pork pie Beef Turkey franks Jellied pork Cooked ham Ox tongue Undercooked chicken	Shrimp salad Tuna salad Smoked fish

**Table 2 microorganisms-12-00178-t002:** Characteristics of patients with *Listeria monocytogenes* vertebral osteomyelitis.

Author	Age	Gender	Co-Morbidities	Clinical Symptoms	Duration of Symptoms	Antibiotic Treatment and Duration	Surgery
Adebolu et al. [93]	60	M	Polymyalgia rheumatica	Back pain	12 months	Ampicillin, IV, 6 weeks Gentamicin, IV, 2 weeks	Yes
Khan et al. [94]	69	M	Prior spinal laminectomy	Back pain	5 months	Ampicillin, IV * Gentamicin, IV *	Yes
Camp et al. [95]	67	M	DM, prior lumbar surgery	Back pain	Unknown	Oxacillin, IV * Streptomycin, IV *	Yes
Chirgwin et al. [96]	57	M	DM, asthma	Fever, back pain	3 weeks	Ampicillin, IV, 6 weeks Tobramycin, IV, 6 weeks	Yes
Aubin et al. [97]	92	M	DM, heart failure, hip arthroplasty	Fever	1 week	Amoxicillin, IV, 6 days Gentamicin, IV, 4 days Trimethoprim-sulfamethoxazole, oral, 12 weeks	Yes
Hasan et al. [98]	63	M	DM, aortic valve replacement	Fever, back pain	2 days	Benzyl penicillin, IV, 6 weeks Rifampicin, oral, 4 weeks Amoxicillin, oral, 18 weeks	Yes
Duarte et al. [99]	65	M	DM	Fever	5 days	Ampicillin, IV, 2 weeks Amoxicillin, oral, 12 weeks	Yes
Al Ohaly et al. [100]	79	M	Hypertension, carotid bypass, repair of AAA	Back pain	3 weeks	Ampicillin, IV, 6 weeks	No

M: male; DM: diabetes mellitus; IV: intravenously; AAA: abdominal aortic aneurysm; * duration not specified.

## Data Availability

Not applicable.

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
