# Peer review of "Clinical Findings of Listeria monocytogenes Infections with a Special Focus on Bone Localizations"

_microorganisms, 2024, doi:10.3390/microorganisms12010178_

Round 1

Reviewer 1 Report

Comments and Suggestions for Authors

Listeria monocytogenes is a dangerous pathogen isolated from many environments. It is estimated that the mortality rate of listeriosis is about 30%. The elderly, pregnant women and newborns are most at risk of contracting the disease. Research topics on this microorganism are important for protecting public health. 

Below are comments to the Authors: 

- Please supplement your paper with figures and tables. For the literature review this is essential 

- Please expand on the following: virulence factors of L. monocytogenes, stress factors vs. L. monocytogenes, food sources including Food of non-animal origin, 

- The authors mentioned ActA - please expand on the issue especially in terms of crossing the blood-brain barrier and the "Trojan horse" strategy,

- please provide mortality statistics for all forms of listeriosis mentioned

- add information on antibiotic resistance of L. monocytogenes and acquisition of antibiotic resistance genes 

Author Response

We thank the reviewer for the comments that significantly improved the quality of our manuscript

- Please supplement your paper with figures and tables. For the literature review this is essential

  • We included in the paper 2 tables

- Please expand on the following: virulence factors of L. monocytogenes, stress factors vs. L. monocytogenes, food sources including Food of non-animal origin

- The authors mentioned ActA - please expand on the issue especially in terms of crossing the blood-brain barrier and the "Trojan horse" strategy,

  • Though these were not main topics of our paper, we appreciate this comment. We included a paragraph in the introduction to shortly describe the virulence factors of L. monocytogenes. Furthermore, we described in the part related to the neurological involvement the mechanisms of crossing blood-brain barrier.

- please provide mortality statistics for all forms of listeriosis mentioned

  • The mortality for each clinical complications has been included

- add information on antibiotic resistance of L. monocytogenes and acquisition of antibiotic resistance genes

  • We thank the reviewer for this comment. However, the discussion of antibiotic treatment and resistance is not the topic of our manuscript. This very interesting and crucial topic should be discussed in a separated review, as it would yield the review too long and it would not allow the paper to be focused on the topic we proposed.

Reviewer 2 Report

Comments and Suggestions for Authors

I read the review carefully and with real pleasure. I congratulate the authors for the completeness of the citations. I have sometimes found small typos that I have marked in the review PDF that I attach to my evaluation. I ask the authors to evaluate it. I have marked in one point my request for moderate in-depth analysis on one of the sources of animal listeriosis. In any case, the review is very well structured, complete, full of bibliographical references and pleasant to read.
It may have been useful to include some tables to summarize the results of studies on clinical cases of human listeriosis, but in any case the review provides the essential updating elements for this specific field of study.

Comments on the Quality of English Language

I found only minor inaccuracies in writing or English. I have pointed out these inaccuracies in the pdf file that I attach to my review.

Author Response

We wish to thank the reviewer for the comments. We modified our paper accordingly. As suggested, we also included two tables in the manuscript.

Round 2

Reviewer 1 Report

Comments and Suggestions for Authors

Please add at least one figure/scheme to the manuscript.

Author Response

Thanks for the comment

We included in the text a figure on the MRI evolution of a patient with epidural abscess (it was part of a case series that we published last year)